# Chemical Characterization of Plant Extracts and Evaluation of their Nematicidal and Phytotoxic Potential

**DOI:** 10.3390/molecules26082216

**Published:** 2021-04-12

**Authors:** Raúl Velasco-Azorsa, Héctor Cruz-Santiago, Ignacio Cid del Prado-Vera, Marco Vinicio Ramirez-Mares, María del Rocío Gutiérrez-Ortiz, Norma Francenia Santos-Sánchez, Raúl Salas-Coronado, Claudia Villanueva-Cañongo, Karla Isabel Lira-de León, Beatriz Hernández-Carlos

**Affiliations:** 1Instituto de Recursos, Universidad del Mar, Puerto Ángel, San Pedro Pochutla, Oaxaca 70902, Mexico; rvazorsa@gmail.com; 2Instituto de Agroindustrias, Universidad Tecnológica de la Mixteca, Acatlima, Huajuapan de León, Oaxaca 69000, Mexico; sacrumix@hotmail.com (H.C.-S.); nsantos@mixteco.utm.mx (N.F.S.-S.); rsalas@mixteco.utm.mx (R.S.-C.); villanueva.vc@gmail.com (C.V.-C.); 3Colegio de Postgraduados, km 36.5 Carretera México-Texcoco, Montecillos, Estado de Mexico, Texcoco 56230, Mexico; icid@colpos.mx; 4Departamento de Ingeniería Química y Bioquímica, Tecnológico Nacional de México/I.T. Morelia, Av. Tecnológico 1500, Lomas de Santiaguito, Morelia 58120, Mexico; jvinicio2000@yahoo.com.mx; 5Instituto de Ecologia, Universidad del Mar, Puerto Ángel, San Pedro Pochutla, Oaxaca 70902, Mexico; rocio@angel.umar.mx; 6Facultad de Química, Universidad Autónoma de Querétaro, Las Campanas, Querétaro 76010, Mexico; liraleonki@gmail.com

**Keywords:** *Nacobbus aberrans*, nematicidal activity, α-terthienyl, stigmasterol, quercetagetin derivatives, 9-*O*-angeloyl-retronecine N-oxide

## Abstract

*Nacobbus aberrans* ranks among the “top ten” plant-parasitic nematodes of phytosanitary importance. It causes significant losses in commercial interest crops in America and is a potential risk in the European Union. The nematicidal and phytotoxic activities of seven plant extracts against *N. aberrans* and *Solanum lycopersicum* were evaluated in vitro, respectively. The chemical nature of three nematicidal extracts (EC_50,48h_ ≤ 113 µg mL^−1^) was studied through NMR analysis. Plant extracts showed nematicidal activity on second-stage juveniles (J2): (≥87%) at 1000 µg mL^−1^ after 72 h, and their EC_50_ values were 71.4–468.1 and 31.5–299.8 µg mL^−1^ after 24 and 48 h, respectively. Extracts with the best nematicidal potential (EC_50,48h_ < 113 µg mL^−1^) were those from *Adenophyllum aurantium*, *Alloispermum integrifolium*, and *Tournefortia densiflora*, which inhibited *L. esculentum* seed growth by 100% at 20 µg mL^−1^. Stigmasterol (**1**), β-sitosterol (**2**), and α-terthienyl (**3**) were identified from *A. aurantium*, while **1**, **2**, lutein (**4**), centaurin (**5**), patuletin-7-β-*O*-glucoside (**6**), pendulin (**7**), and penduletin (**8**) were identified from *A. integrifolium*. From *T. densiflora* extract, allantoin (**9**), 9-*O*-angeloyl-retronecine (**10**), and its N-oxide (**11**) were identified. The present research is the first to report the effect of *T. densiflora*, *A. integrifolium*, and *A. aurantium* against *N. aberrans* and chemically characterized nematicidal extracts that may provide alternative sources of botanical nematicides.

## 1. Introduction

The plant parasite nematode, *Nacobbus aberrans* (Thorne, 1935; Thorne & Allen, 1994), also known as false root-knot, is distributed in Argentina, Bolivia, Chile, Ecuador, Mexico, Peru, and the USA. It reduces crop yield in tomatoes (*Solanum lycopersicum* L.), potatoes (*Solanum tuberosum*), beans (*Phaseolus vulgaris* L.), and sugar beets (*Beta vulgaris*). The nematode ranks among the “top ten” plant-parasitic nematodes of phytosanitary importance [1]. It is estimated it meets the criteria to be a potential risk in the EU [2]. Management strategies of plant-parasitic nematodes are cultural practices (crop rotation), mixed-cropping, organic amendments, resistant crop cultivars, biological control [3,4,5,6], chemical nematicides, and bioactive products of plant origin. Among these strategies, natural product usage represents a vital option for controlling phytopathogenic nematodes because of their low impact on the environment and non-target organisms. In the search for botanic nematicides, some of the most recent proposals are using *Stevia rebaudiana* and *Origanum vulgare* to control *Meloidogyne*; in vivo experiments showed this effect [7,8]. In the case of *N. aberrans*, crude herbal extracts from *T. lunulate* [9], *Cosmos sulphureus* [6], *Senecio salignus* [6], *Witheringia stramoniifolia* [6], and *Lantana cámara* [6] showed in vitro nematicidal activity at 500 µg mL^−1^ (>70%) to second-stage juvenile (J2) individuals. Simultaneously, in vivo protection from infection of *Lycopersicum esculentum* Mill and *Capsicum annumm* plants occurred with extracts of *Tagetes erecta* [9] and *Trichilia galuca* [10], respectively. There are only two reports of natural compounds with toxic potential for the control of *N. aberrans*: capsidiol acts as a nematostatic on *N. aberrans* J2 (90% immobility) at 1.5 µg mL^−1^ after 72 h [11]; and various cadinenes affect immobility-mortality (LC_50_ 25.4–111.4 µg mL^−1^) and inhibit eggs hatching (IC_50_ 31.23–56.71 µg mL^−1^) [12]. Identification of substances from botanical origins capable of controlling *N. aberrans* lies partially on differences with other common plant-parasitic nematodes (e.g., *Meloidogyne* spp.), such as its endoparasitic stage (migratory and sedentary), several infective stages, and 15–30 days dehydration tolerance of J3 and J4 larvae [13]. This research identifies botanical sources of compounds with nematicidal or nematostatic characteristics by screening seven plant species for nematicidal activities against *N. aberrans* J2 individuals and phytotoxicity to tomato seeds. NMR analysis established the chemical nature of nematicidal extracts (EC_50,48h_ ≤ 112.3 µg mL^−1^) as sterol, thiophene, flavonoid, and alkaloid-like compounds (Figure 1). Some of these compounds are known for their nematicidal or nematostatic activity against other nematode species.

## 2. Results and Discussion

### 2.1. Nematostatic and Nematicidal Effect and EC_50_ of Extracts

From the nine extracts evaluated in this study (Table 1), *G. mexicanum* and *A. integrifolium* achieved the maximum paralysis effect (immobility = 85.3 ± 10% and 94.1 ± 3%, respectively) at 1000 µg mL^−1^ after 24 h. The remaining extracts showed their highest nematostatic effects after 36–60 h (immobility = 91.3%–100%). The immobility percentage after 72 h reached >95% for all extracts. Nematicidal (72 h *) and nematostatic effects (72 h) at 1000 µg mL^−1^ occurred at similar immobility percentages for A. aurantium (roots), A. integrifolium, A. subviscida, G. mexicanum, and H. terebinthinaceus, while the rest of the extracts showed their nematicidal effects at lower immobility percentages due to recovery of mobility (6.3–11%) (Table 1).

Extracts at 100 µg mL^−1^ produced the highest immobility rates after 24 h (*A. aurantium* A), 36 h (*A. integrifolium*), 48 h (*A. aurantium* R), 60 h (*A. cuspidata* and *G. mexicanum*), and 72 h (*A. subviscida*, *H. terebinthinaceus*, and *T. densiflora* R and A) with immobility percentages ranging from 54.0 to 73.6 (Table 2). Lower concentration extracts (10 µg mL^−1^) produced immobility percentages <26% (Appendix A). Table 3 shows calculated EC_50_ values that cause 50% immobility in *N. aberrans* J2 individuals at 24 and 48 h. EC_50_ values for 24 h ranged from 53.5 to 468.1 µg mL^−1^ (ci, α = 0.05) and for 48 h varied from 31.5 to 299.8 µg mL^−1^. The most effective extracts (EC_50,48h_ ≤ 112.3 µg mL^−1^) were *A. aurantium* R (62.3–88.3 µg mL^−1^), *A. aurantium* A (31.5–110.4 µg mL^−1^), *A. integrifolium* (47.4–107.1 µg mL^−1^), and *T. densiflora* R (59.0–112.3 µg mL^−1^). Chemical, non-fumigant nematicides acted as positive controls (Fluopyram and Abamectin) with EC_50_ values of 27.8–31.3 and 25.0–27.4 µg mL^−1^ at 24 and 48 h, respectively. Fluopyram is a succinate dehydrogenase inhibitor that blocks cellular respiration, while abamectin acts on glutamate-gated chloride channels, causing nematode death [14].

### 2.2. Compounds Identified in A. aurantium and their Nematostatic Effects

Comparison of ^1^H and ^13^C NMR data allowed identification of stigmasterol (**1**) and β-sitosterol (**2**) [15,16]. Previously, we described the isolation and identification of α-terthienyl (**3**) and 5-(4″-hydroxy-1′-butynyl)-2–2′-bithiophene from *A. aurantium* roots [17]. In the present research, nematicidal extracts against *N. aberrans* contained 3.

Several treatments were tested at 50 and 100 µg mL^−1^: pure compounds **1** and **3** (Figure 1); a mixture of 1 and 2 (Mixt); α-terthienyl (aT); stigmasterol (St); and β-sitosterol (bS) (Figure 2). At 100 µg mL^−1^, **1**, **3**, and Mixt reached maximum inhibition at 72 h (93.3 ± 3%), 36 h (99.4 ± 0.6%), and 24 h (88.3 ± 8%), respectively. Commercial compounds, β-sitosterol, α-terthienyl, and stigmasterol showed lower nematostatic effects (50.6–80.7%), mainly during the first few hours of observation (12–36 h) (Figure 2). After 72 h, isolated and commercial β-stigmasterol showed 100 ± 0.0 and 94.5 ± 5.3% nematostatic effects, respectively. Treatments of isolated and commercial α-terthienyl showed the same immobility percentages (α = 0.01) after 72 h with values of 93.3 ± 3.1 and 90.6 ± 5%, respectively. Commercial α-terthienyl, stigmasterol, and β-sitosterol showed nematodes recovery values of 0.83, 0.61, and 0.61%, respectively (Table 4). Therefore, these compounds are nematostatic and nematicides at the same concentrations (α = 0.05).

Thiophenes are common in the *Tagetes* genera [18,19]; this is a cover crop against plant-parasitic nematodes *Meloidogyne* and *Pratilenchus* (*T. erecta*, *T. patula*, *T. tenuifolia*, and *T. minuta*) [20]. Some thiophenes described as nematicide are 1-phenylhepta-1,3,5-triyne, and 5-phenyl-2-(1′-propynyl)-thiophene. These thiophenes, isolated from *Coreopsis lanceolata* L., effectively treated the pinewood nematode *Bursaphelenchus xylophillus* at 2 mM [21]. A commercial sample of α-terthienyl caused 100% mortality after 24 h at 0.125% against the infective larval stage of the cyst nematode *Heterodera zeae* [22]. In our experiments, α-terthienyl showed 83.2 ± 5.2% immobility after 24 h at 100 µg mL^−1^, or 0.01% against *N. aberrans*, while the commercial compound showed less effectiveness (71.74 ± 8.8%) (Figure 2). Sterols functioned on nematicidal activity as has been documented: β-sitosterol showed 60% mortality in *M. incognita* at 1% (after 12 h) [23] and together with stigmasterol at 5 µg mL^−1^ caused 74.4% and 55.3% mortality in *M. incognita* and *Heterodera glycines*, respectively [24]. In this work, isolated stigmasterol showed maximum activity after 36 h (99.4 ± 0.56%), while the commercial compound achieved it after 72 h (94.5 ± 5.3%). Synergy from impurities in the natural stigmasterol could account for the observed stronger activity. The impurities were β-sitosterol and a very similar compound but with 2 oxygen atoms, which was not identified. Similar results caused by β-sitosterol were observed with increased immobility from 68.7 ± 8.5% to 75.8 ± 12.5% when stigmasterol is present (Figure 2). A recent report about synergistic effects discusses nematicidal activity against *Meloidogyne incognita* of a mixture of 23a-homostigmast-5-en-3β-ol and nonacosan-10-ol. The mixture, at 50 µg mL^−1^, showed 93.7% mortality after 24 h, while the compounds individually, at 100 µg mL^−1^, exerted 50% mortality (24 h) [25,26]. The sterol kill mechanism on nematodes may disrupt steroid metabolism as stigmasterol (**1**) possesses a chemical similarity to α-ecdysone (Figure 1). α-ecdysone is involved in the biosynthesis and metabolism of molting and sex hormones of nematodes [27]. Such a function took part in accelerating the development of *M. incognita* by applying an ecdysone derivate (0.5 mM) on tomato seeds [28].

### 2.3. Compounds Identified in A. integrifolium

Structural identification of 1, 2, 4–8 was made by comparing their ^1^H and ^13^C NMR data with those described [15,16,29,30,31,32,33,34]. Identification of 5–8 required HSQC and HMBC experiments to confirm the structures and assign ^1^H and ^13^C signals (Appendix A).

Previous studies of *A. integrifolium* include the presence of an acetylene compound from stems [35]. Additionally, methanol extracts from stems showed inhibition of a topoisomerase enzyme (JN394, −81.19 ± 2.12% and JN362a, 126.06 ± 12.02%), and no significant antioxidant (AE = 4.18%) nor antimicrobial activities (MIC ≥ 6250 µg mL^−1^) [36]. In this research, the identification of quercetagetin derivatives 5–8 in *A. integrifolium* matches the genus’s phytochemistry (also named *Calea* genus) [37]. Previous studies have shown the nematicidal potential of flavonoids. For example, rutin exhibited the same mortality percentage as the positive control (carbofuran): 100% mortality at a 0.5% concentration (24 h) against the cyst nematode *Heterodera zeae*. Moreover, quercetin showed 50% mortality, and patuletin 7-β-*O*-glycoside showed no significant activity (20%, 24 h, light) [22]. However, this compound exerts moderate activity against *Meloidogyne incognita* J2 individuals (LC_50,48h_ = 0.506%), while kaempferol, isorhamnetin, rutin, myricetin, and fisetin, among others, exhibited an LC_50,48h_ value similar to carbofuran (LC_50,48h_ = 0.0506%) [38]. Nematicidal activity of flavonoids may be due to acetylcholinesterase inhibition (AChE) because nematodes possess some neurotransmitters common in mammals like acetylcholine, serotonin, or glutamate [39]. Flavonoids as quercetin, genistein, and luteolin 7-*O*-glycoside inhibited AChE by 76.2, 65.7, and 54.9%, respectively [40], while a methoxylated quercetagetin showed a minor effect (inhibition 23.73 ± 1.94%) [41]. However, a glycosylated derivative, quercetagetin-7-*O*-(6-*O*-caffeoyl-β-D-glucopyranoside, possessed significant activity (IC_50_ 12.54 ± 0.50 µg mL^−1^) against AChE isolated from *Caenorhabditis elegans* and *Spodoptera litura*. The IC_50_ value found came close to the value for chloropyrifos (2.32 ± 0.06 µg mL^−1^) [42]. We hypothesize flavonoid (**5**–**8**) isolated from *A. integrifolium* exerted their effects on the cholinergic nervous system of *N. aberrans* J2 individuals, while sterols (triterpenes) (**1**, **2**) acted on the hormonal system. Therefore, the nematostatic (EC_50,48h_ = 47.4–107.1 µg mL^−1^) and nematicidal (at 1000 µg mL^−1^, 99.4 ± 1%) effects observed could be the synergetic interaction between sterols and flavonoids. Synergic effects have been described on triterpenes (as saponins) and polyphenols combinations, which improved nematicidal activity in vivo against various nematode species (*Meloidogyne*, *Xiphinema*, *Tylenchorhynchus*, *Criconemoides*, and *Pratylenchus*) [43].

### 2.4. Compounds Identified in T. densiflora

Studies of the chemical composition of extracts from the *Tournefortia* genera revealed phenolic compounds [44] and alkaloids [45]. We identified allantoin (**9**) and pyrrolizidine alkaloids **10** and **11** (PAs), initially identified in the methanolic extract from its NMR data. The ^13^C NMR spectrum showed signals at δ 96.70, 96.28, and 96.21 (Figure 3) possibly related to C-8 of N-oxide PAs [46,47] and signals at δ 76.74 and 76.66 probably due to C-8 of PAs [48]. The detection of PAs in *T. densiflora* roots required alkaloid extraction and chromatographic separation to identify **10** and **11**. ^1^H and ^13^C NMR data comparison with similar data from retronecine and PAs N-oxides [48,49,50,51], as well as HMQC and HMBC experiments, identified 9-*O*-angeloyl-retronecine N-oxide (**11**). The DEPTQ spectrum revealed the following functional groups: two CH_3_, four CH_2_, four CH, and three quaternary carbon atoms. Two double bonds were observed from signals at δ 132.9 (C), 121.4 (CH), 127.6 (C), and 139.0 (CH), while an angeloyl ester moiety was deduced from the ^13^C signal at δ 167.1 and the ^1^H NMR signal at δ 6.16 as qq (*J* = 1.6, 8.4Hz, H-12), which showed HMBC correlation with signals at δ 20.1 (C-12) and 16.0 (C-13). The signal at δ 95.5, caused by CH (C-8) next to quaternary nitrogen (N-oxide), and the signals at δ 34.8 (C-6), 68.5 (C-5), 77.8 (C-3), 60.9 (C-9), and 69.8 (C-7) were consistent with a necine base (as N-oxide) carrying a double-bond between C1 (132.9) and C2 (121.4). HMBC correlations support this proposal, and the presumed position of the angeloyl group at C-9, as the correlation of H-9 (δ 4.78 and 4.72) with the ester carbon C-10 (δ 167.1) showed (Table 5). Compound **10** displayed a ^13^C NMR spectrum very similar to **11**. The main differences lay in the chemical shifts of C-3, C-5, and C-8 (atoms near the quaternary nitrogen), which appear in **10** at 60.4, 53.4, and 77.8 ppm, respectively. In **11**, these signals shifted to higher frequencies at δ 77.8 (C-3), 68.5 (C-5), and 95.5 (C-8) ppm (Table 5).

Previous research demonstrated toxicity to *Saccharomyces cerevisae* from methanolic extracts from roots (70 mgmL^−1^) [52] and mycelial growth inhibition of *Alternaria alternata* (69.07 ± 2.0%) and *Fusarium solani* (52.42 ± 2.0%) [46]. Pyrrolizidine alkaloids (PAs) play a defensive role in the plant as antifeedants against herbivores [53]. Research shows PAs and PAs N-oxides are nematicides against *Meloidogyne incognita*, *Heterodera schachtii*, *Pratylenchus penetrans*, *Plasmarhabditis hermaphrodita*, and *Rhabditis* sp. [54]. Additionally, allantoin (**9**) possesses nematicidal activity (51.3% mortality) in *Meloidogyne incognita* and *Heterodera glycines* at 5 µg mL^−1^ [24]. These toxic activities and common occurrence of PAs in the *Boraginaceae* family [45] confirmed that PAs and allantoin are responsible for the toxic effects of the *T. densiflora* root extract observed on *N. aberrans* J2 individuals. PAs toxicity relies on their oxidation to pyrrolizinium ions by the cytochrome-P450 enzyme in the nematode. These ions are highly reactive as electrophiles and react with DNA, proteins, and other important macromolecules [55].

### 2.5. Phytotoxicity Test

Most nematicidal extracts against *N. aberrans* extracts showed 100% inhibition (−100%) of *L. esculentum* radicle growth at 20 µg mL^−1^, except extracts from *H. terenbinthinaceus* (−37%) (Figure 4). *T. densiflora* R extracts were the most phytotoxic with 40 and 38% inhibition at 0.02 µg mL^−1^ (Figure 4). Potentially, these extracts could act as soil disinfection agents. Also, *A. cuspidata*, *A. subviscida*, and *T. densiflora* A extracts showed hormetic effects: a 20 µg mL^−1^ solution inhibited radicle growth inhibition while a 0.02 µg mL^−1^ solution promoted it.

Nematostatic and nematicidal effects observed by treatments with EC_50,48h_ < 113 µg mL^−1^ relate to secondary metabolites like sterols, flavonoids, thiophenes, or alkaloids (PAs and allantoin), possibly biosynthesized in plants as a stress response. In general, sterols are involved in plants’ growth and fertility as hormonal precursors and cell membranes’ functional components. Nematodes also need sterols for their survival, but they cannot biosynthesize them de novo, so the nematodes readily absorb sterols. For example, *Meloidogyne arenaria*, *M. incognita*, and *Pratilenchus agilis* incorporate and transform sterols into necessary derivatives in their growth and reproduction [56]. Thus, nematodes should elicit a biological response to some sterols. Also, plant–nematode interactions require flavonoids and might be required for nematode reproduction. However, some flavonoids with specific structural arrangements have shown toxic effects on specific targets such as enzymes. Finally, thiophenes could inhibit enzymes like superoxide dismutase [57] and damage DNA [58]. The transformation of secondary metabolites to more toxic compounds also happened with PAs, as mentioned before.

## 3. Materials and Methods

### 3.1. General Experimental Procedures

NMR measurements were carried out on Bruker ASCEND^TM^ 400 (400 MHz proton frequency) spectrometer (Bruker, Germany) at 298 K using 5 mm probes at 22 °C from CD_3_OD or DMSOd_6_ solutions. Chemical shifts (δ = ppm) were referenced to 2.50 (^1^H) and 39.43 (^13^C) ppm (DMSOd_6_) or to 3.30 (^1^H) and 36.067 (^13^C) ppm (CD_3_OD). Coupling constants are given in Hz. Signals are described as s (singlet), d (double), t (triple), and q (quartet).

### 3.2. Chemicals

All reagents and solvents (ACS grade), LiChroprep RP-18, and SiO_2_ supports for column and plate chromatography were obtained from Merck (MA, USA). Amberlite XAD16, α-terthienyl, β-sitosterol, stigmasterol, deuterated solvents, and dimethyl sulfoxide (DMSO-Hybri-Max) were obtained from Sigma Chemical (St. Louis, MO, USA).

### 3.3. Plant Species

The plant species were collected in Oaxaca, Mexico (See Table 6), and voucher specimens were deposited in the Herbarium of Forest Sciences, Universidad Autonoma de Chapingo, Texcoco (Estado de México, México). The scientific name, collection site, voucher number, plant part used, and extraction solvent are listed in Table 6.

### 3.4. Preparation of Extracts

Extracts were prepared according to procedures previously described [52]. All extracts were kept at 4 °C and protected from light and moisture until further use.

### 3.5. Isolation of ***1**–**3*** from A. aurantium Extract

The methanol extract (0.582 g) from aerial parts of *A. aurantium* was subjected to column chromatography (CC) using *n*-hexane-ethyl acetate mixtures. The fractions eluted with an 8:2 mixture, were re-chromatographed on CC with *n*-hexane-ethyl acetate (95:5) to yield 63.2 mg (10.8%), 30 mg (5.15%), and 3.02 mg (0.52%) of stigmasterol (**1**) and a mixture of stigmasterol (**1**)/β-sitosterol (**2**), and α-terthienyl (**3**), respectively (Figure 1). The purity of stigmasterol and α-terthienyl was approximately 95 and 98%, respectively. Purity was approximate since ^1^H NMR spectra by comparison of integration areas of **1** and **3** with those corresponding to impurities. The methanol extract from roots (7.215 g) was dissolved in acetone, and the solution yielded a solid residue (0.347 g), which was subjected to CC using *n*-hexane-ethyl acetate mixtures to obtain 40 mg (0.55%) and 23.7 mg (0.32%) of **1** and **3** respectively.

### 3.6. Identification of Compounds from A. integrifolium

Methanol extract of *A. integrifolium* (23.1 g) was partitioned with ethyl acetate (3 times) to obtain 10.1 g of ethyl acetate soluble fraction (ESF) and 13 g of methanol soluble fraction (MSF). ESF was subjected to column chromatography (SiO_2_) and eluted with mixtures of AcOEt: *n*-hexanes to obtain a mixture of chlorophylls “a” and “b” (80 mg), and a dark solid (155.8 mg). The solid was re-chromatographed (SiO_2_) with the same eluents to obtain lutein (**4**, 9.3 mg) (Figure 1) and a mixture (27.1 mg) of stigmasterol (**1**) and β-sitosterol (**2**). MFS (13 g) was solubilized in water and supported on a column of Amberlite XAD16; after two washes with water, the compounds retained were eluted with methanol to obtain a residue (1.5 g) free from simple carbohydrates. The residue (1.0 g) was eluted in a chromatography column (C_18_) using methanol:water mixtures as eluent. Chromatographic separation yielded 72 mg of four quercetagetin derivatives in binaries mixtures; its approximate composition was calculated by integrating ^1^H NMR areas of their characteristic signals. These compounds were identified as centaurin (**5**, 28.1 mg), patuletin-7-β-*O*-glucoside (**6**, 1.7 mg), pendulin (**7**, 6.4 mg), and penduletin (**8**, 1.0 mg) (Figure 1) from the analysis of their NMR data (Appendix A).

### 3.7. Identification of Compounds from T. densiflora Roots

The methanol extract (1.14 g), previously defatted with *n*-hexane and AcOEt, was subjected to C_18_ column chromatography and eluted with water. The eluent was identified by its NMR data [59] as allantoin (9, 33 mg, Figure 1, Table 5). PAs extraction required 8 g of the methanolic extract to be stirred with 1 M HCl (44 mL, 20 min); the mixture was filtered, the filtrate adjusted to pH 10 (KOH 1 M), and extracted with ethyl acetate. Organic fraction (180 mg) was subjected to chromatographic column (CC-SiO_2_) and eluted with CHCl_3_: methanol mixtures to obtain 9-*O*-angeloyl-retronecine (**10**, 11.5 mg, approx. 80% purity) and their N-oxide (**11**, 12.1 mg, approx. 90% purity).

### 3.8. Screening of Nematicidal and Nematostatic Activities

#### 3.8.1. Nematodes

Mature egg masses of *N. aberrans* were extracted from infected roots of tomato plants (*Lycopersicum esculentum* Mill., 1768 or *Solanum lycopersicum*), propagated at Colegio de Postgraduados, Montecillo, Texcoco, Mexico. Egg masses were gently washed with water to remove adhered soil and a NaOCl 0.53% solution until the gelatinous matrix dissolved. Then they were washed with distilled water on a mesh sieve (#400) and incubated in distilled water at 25 °C for 5 days. Emerging J2 individuals were used in all experiments.

#### 3.8.2. Assay

Test solutions were prepared in DMSO with 0.5% Tween 20 at 10, 100, 1000 µg mL^−1^ for extracts, while concentrations at 100 µg mL^−1^ were used for compounds and fractions. Fluopyram 50% (Verango, Bayer) and abamectin 5.41% (Oregon 60C-FMC) at 5, 10, 15, 25, 30 and 50 µg mL^−1^ (dissolved in distilled water) were tested as positive control. Treatments (5 µL) and between 100 and 150 J_2_ individuals in 95 µL of water were added to 96-well plates (Falcon, USA) and incubated at 25 °C. DMSO with 0.5% Tween 20 (5 µL) in 95 µL of water was used as blank. Previously, non-effect on J2 mobility was shown at 24, 36, 48, 60, and 72 h with the solvents used (Appendix A). Percentages of J2 immobility were recorded after 12, 24, 36, 48, 60, and 72 h by counting mobile and immobile J_2_ individuals under a stereomicroscope at 240X. A nematode was considered immobile if the nematode failed to respond to stimulation with a needle. After that, the J_2_ individuals at 1000 µg mL^−1^ (extracts) and 100 µg mL^−1^ (commercial stigmasterol, α-terthienyl, and β-sitosterol) were washed on a 400-mesh filter with distilled water to remove the excess test substance (extracts and commercial compounds). The treatments were replaced with distilled water to allow a possible recovery of the J2 individuals after 24 h. If they remained immobile, they were assumed to be dead, and the effect was considered nematicide. If any J_2_ individual regained mobility, the effect was considered nematostatic (paralysis). All treatments (extracts, isolated and commercial compounds) and control were replicated five times, and the experiments were performed two times. The immobility percentage was calculated using the equation: i = 100 × (1 − n_t_/n_c_); where i = immobility percentage, n_t_ = active J_2_ in the treatment, and n_c_ = active J_2_ in the blank [60].

### 3.9. Phytotoxicity Assay

Experiments were conducted with *L. esculentum* F1 seeds var. Sheva according to the methodology described [61]. Prior to evaluation, all extracts were dissolved in a 0.5% DMSO/H_2_O solution at 20, 2.0, 0.2, 0.02, and 0.002 µg mL^−1^ concentrations to obtain solids-free solutions. Commercial herbicide (Glyphosate) was used as a positive control at the same concentrations, while 0.5% DMSO/H_2_O was used as blank (100% growth).

### 3.10. Statistical Analysis

All experimental data were subjected to an analysis of variance (ANOVA) using Statistica Pro (Stat Soft, Japan). Treatment means were tested with Tukey’s HSD multiple comparison test at 0.05% or 0.01% probability levels.

## 4. Conclusions

To our knowledge, our results show for the first time the nematicidal activity against *N. aberrans* from *T. densiflora*, *A. integrifolium*, and *A. aurantium* extracts. In this research, we identified several compounds present in the nematicidal extracts against J2 individuals of *N. aberrans* containing: (a) flavonoids (*A. integrifolium*); (b) triterpene-type compounds (*A. aurantium*, *A. integrifolium*), (c) thiophene-type compounds (*A. aurantium*) and (d) alkaloids (*T. densiflora*). We identify **5**–**8** and **9**–**10** from *A. integrifolium* and *T. densiflora*, respectively. Moreover, we described the phytotoxic effect of all extracts on tomato radicle growth. Further research of these plant extracts will allow us to identify more compounds responsible for the nematicidal activity and provide alternative nontoxic crop protection chemicals.

## Figures and Tables

**Figure 1 molecules-26-02216-f001:**
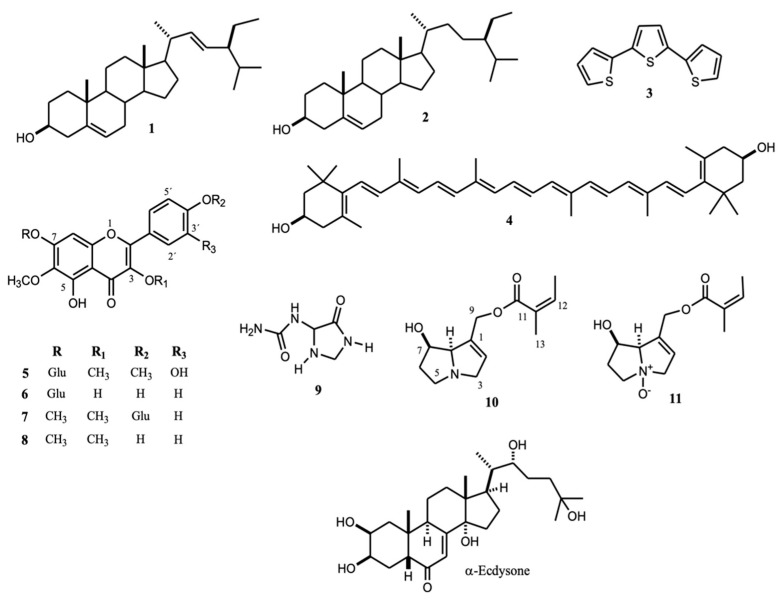
Structure of compounds identified from nematicidal extracts against J_2_ individuals of *N. aberrans* (**1**–**11**) and α-ecdysone structure.

**Figure 2 molecules-26-02216-f002:**
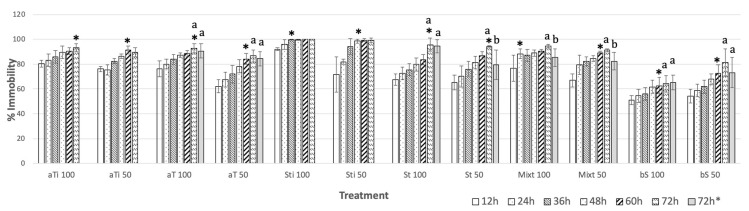
Effect of compounds isolated α-terthienyl (aTi), stigmasterol (Sti), a mixture of 1 and 2 (Mixt), and commercial compounds: stigmasterol (St), α-terthienyl (aT) and β-sitosterol (bS) on the immobility of *N. aberrans* J_2_ individuals. * Maximum percentage of immobility. According to Tukey’s test, columns followed by the same letter are not significantly different (*p* < 0.05). Concentration 100 and 50 µg mL^−1^.

**Figure 3 molecules-26-02216-f003:**
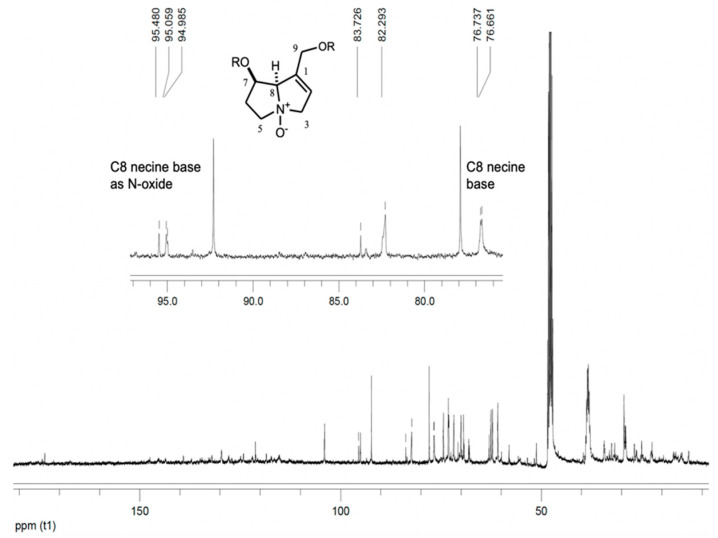
^13^C NMR spectrum of methanol extract of *T. densiflora* roots. 100 MHz, CD_3_OD.

**Figure 4 molecules-26-02216-f004:**
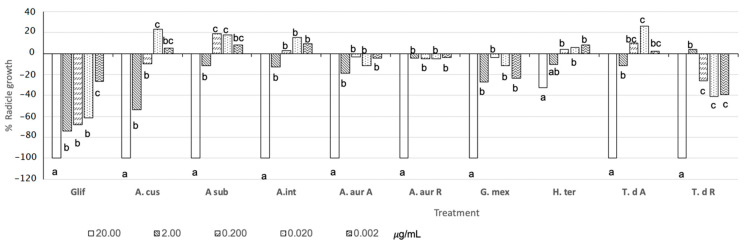
Phytotoxic activity on *L. esculentum* of plant extracts.

**Table 1 molecules-26-02216-t001:** Effect of plant extracts at 1000 µg mL^−1^ on the immobility of *N. aberrans* J2s individual (100–150) after different exposure times.

Extract	% Immobility J2s						
	12 h	24 h	36 h	48 h	60 h	72 h	72 h *
*A. aurantium* A	84.6 ± 6 a	89.32 ± 1 b	97.0 ± 2 bc	98.1 ± 3 bc	99.3 ± 1.1 c	99.2 ± 1.5 c	96.4 ± 5.4 bc
*A. aurantium* R	61.0 ± 16 a	65.97 ± 6 a	95.3 ± 4 b	99.0 ± 0 b	100 ± 0 b	100 ± 0.0 b	99.8 ± 0.5 b
*A. cuspidata*	82.3 ± 5 a	86.6 ± 4 a	94.6 ± 3 ab	96.0 ± 1 b	98.3 ± 2 b	98.9 ± 2 b	90.6 ± 5 a
*A. integrifolium*	61.8 ± 13 a	94.1 ± 3 b	93.3 ± 4 b	96.5 ± 2 b	99.6 ± 1 b	99.8 ± 0.4 b	99.4 ± 1 b
*A. subviscida*	78.9 ± 10 a	79.3 ± 10 a	91.3 ± 6 b	93.5 ± 6 b	98.1 ± 2 b	99.3 ± 1 b	95.4 ± 2 b
*G. mexicanum*	62.9 ± 23 a	85.3 ± 10 b	93.7 ± 3 b	96.2 ± 3 b	98.5 ± 2 b	98.6 ± 2 b	94.2 ± 3 b
*H. terebinthinaceous*	86.5 ± 7 a	91.0 ± 16 ab	99.2 ± 1.2 b	97.5 ± 4 ab	99.6 ± 0.8 b	96.9 ± 11 ab	87 ± 11 ab
*T. densiflora* R	80.3 ± 6 a	87.7 ± 13 ab	92.9 ± 2 bc	96.3 ± 2 c	98.1 ± 3 c	98.8 ± 1 c	92.5 ± 3 bc
*T. densiflora* A	79.0 ± 9 a	85.9 ± 9 a	93.2 ± 3 ab	98.9 ± 2 b	98.1 ± 2 b	95.6 ± 3.2 b	88.0 ± 3 ab

* Immobility after 24 h. A: Stem part, R: roots. Data shown correspond to the average of all values ± sd. According to Tukey’s test, the same letter indicates data in each row is not significantly different (*p* < 0.05).

**Table 2 molecules-26-02216-t002:** Effect of plant fractions and extracts at 100 µg mL^−1^ on the immobility of *N. aberrans* J2s individuals (100–150) after different exposure times.

Extract	% Immobility J2s					
	12 h	24 h	36 h	48 h	60 h	72 h
*A. aurantium* A	44.1 ± 6 a	54.02 ± 7 b	68.8 ± 5 b	68.3± 6 b	73.8 ± 6.1 ab	70.8 ± 10 b
*A. aurantium* R	40.1 ± 14 a	34.1 ± 6 ab	52.1 ± 6 b	63.1 ± 4 c	70.2 ± 6 c	71.3 ± 7 c
*A. cuspidata*	44.3 ± 15 b	29.3 ± 7 a	19.6 ± 9 a	25.2 ± 10 a	64.2 ± 5 c	73.7 ± 4 c
*A. integrifolium*	17.6 ± 8 a	42.0 ± 14 b	62.31 ± 11 c	75.3 ± 4 c	68.0 ± 10 c	72.7 ± 10 c
*A. subviscida*	31.4 ± 13 ab	46.0 ± 10 abc	29.6 ± 19 ab	24.50 ± 4 a	44.4 ± 26 bc	60.2 ± 17 c
*G. mexicanum*	20.1 ± 8 a	28.8 ± 5 a	26.4 ± 8 a	40.4 ± 6 b	58.4 ± 11 c	66.9 ± 9 c
*H. terebinthinaceous*	31.1 ± 10 a	62.9 ± 8 bc	62.9 ± 8 bc	61.0 ± 5 b	69.4 ± 6 bc	73.6 ± 3 c
*T. densiflora* R	46.5 ± 20 bc	23.2 ± 12 a	33.0 ± 8 ab	49.1 ± 6 cd	63.2 ± 8 de	66.7 ± 10 e
*T. densiflora* A	45.9 ± 9 a	35.7 ± 16 a	36.5 ± 15 a	37.0 ± 11 a	52.7 ± 17 ab	64.6 ± 13 b

Data shown correspond to the average of all values ± sd. According to Tukey’s test, the same letter indicates data in each row is not significantly different (*p* < 0.05).

**Table 3 molecules-26-02216-t003:** Effective concentrations (50% immobility) at 24 and 48 h on *N. aberrans*.

Extract	EC_50,24h_ µg mL^−1^	EC_50,48h_ µg mL^−1^
*A. aurantium A*	53.5–187.4	31.5–110.4
*A. aurantium R*	289.4–468.1	63.2–88.3
*A. cuspidata*	132.1–282.8	54.1–199.3
*A. integrifolium*	71.4–214.1	47.4–107.1
*A. subviscida*	124.7–342.2	60.0–299.8
*G. mexicanum*	93.0–319.6	74.4–183.4
*H. terebinthinaceous*	84.5–183.4	56.0–164.4
*T. densiflora R*	170.3–301.8	59.0–112.3
*T. densiflora A*	91.3–184.7	81.9–166.1
Fluopyram	27.8–28.4	25.0–26.4
Abamectin	30.6–31.3	26.9–27.4

Confidence limit (α = 0.05).

**Table 4 molecules-26-02216-t004:** Effect of mixture stigmasterol/β-sitosterol, commercial and isolated compounds from *A. aurantium* roots on the immobility of *N. aberrans* J_2_ individuals after 72 h.

Treatment	Concentration µg mL^−1^	% Immobility	% Immobility after Washing
β-sitosterol	100 †	68.7 ± 8.5 a	68.12 ± 8.0
stigmasterol/β-sitosterol	100	88.3 ± 8.1 b	---
α-terthienyl	100	93.3 ± 3.1 bc	--
	100 †	90.6 ± 5.60 bc	89.72 ± 5.5
stigmasterol	100	100.0 ± 0.0 c	--
	100 †	94.5 ± 5.3 bc	93.85 ± 5.6

Data shown correspond to the average of all values ± sd. According to Tukey’s test, the same letter indicates data in each row is not significantly different (*p* < 0.01). † Commercial compounds.

**Table 5 molecules-26-02216-t005:** ^1^H (400 MHz) and ^13^C NMR (100 MHz) data for **9** (MeOD), **10** and **11** (DMSO-*d*_6_).

Atom	9		10		11	
	δ ^1^H	δ ^13^C	δ ^1^H	δ ^13^C	δ ^1^H	δ ^13^C
1	10.54 s	-		133.5	-	132.9
2	-	157.8	5.8 d (1.3 Hz)	122.9	5.80 bs	121.4
3	8.05 s	-	4.15 d (14.7 Hz)	60.4	4.28 d (16.0 Hz)	77.8
			3.77 d (14.7 Hz)		4.56 so	
4	-	174.1	-	-	-	-
5	5.24 d (8.0)	62.9	3.61 so	53.4	3.71 so	68.5
			3.09 ddd (6.4 Hz)		3.60 so	
6	6.89 d (8.0)	-	1.98 so	35.8	2.45 bs	34.8
			1.98 so		1.92 so	
7	-	157.2	4.43 bs	68.9	4.57 so	69.8
8	5.79 s	-	4.61 bs	77.8	4.57 so	95.5
9	-	-	4.78 bs 4.78 bs	60.3	4.78 d (14 Hz) 4.72 d (14 Hz)	60.9
10	-	-	-	166.9	-	167.1
11	-	-	-	127.2	-	127.6
12	-	-	6.18 qq (1.3, 7.3)	138.3	6.16 qq (1.6, 8.4 Hz)	139.0
13	-	-	1.87 q (1.3)	20.5	1.87 q (1.6 Hz)	20.1
14	-	-	1.93 dq (1.3, 7.3)	15.9	1.95 dq (1.6, 8.4 Hz)	16.0

**Table 6 molecules-26-02216-t006:** Plants used in experiments.

Specie (Family)	Collection Site	Voucher Number	Part Plant Used	Extraction Solvent
*Acalypha cuspidata* Jacq. (Euphorbiaceae)	B	25068	Stem	MeOH
*Acalypha subviscida* S. Watson var. Lovelanddii McVaugh (Euphorbiaceae)	A	24007	Stem	MeOH
*Alloispermum integrifolium* (DC.) H. Rob. (Asteraceae)	A	24024	Stem	MeOH
*Adenophyllum aurantium* (L.) Strother (Asteraceae)	C	25173	Stem Root	MeOH MeOH
*Galium mexicanum* Kunth (Rubiaceae)	A	23994	Stem	MeOH
*Heliocarpus terebinthinaceus* (DC.) Hochr. (Tiliaceae)	D	25225	Seeds	H_2_O
*Tournefortia densiflora* M. Martens & Galeotti (Boraginaceae)	C	25221	Stem Root	MeOH MeOH

MeOH: methanol. A: San Miguel Suchixtepec, Miahuatlán; B: Candelaria Loxicha, San Pedro Pochutla. C: Chepilme Garden (Universidad del Mar), SanPedro Pochutla, D: Huajuapan.

## Data Availability

Not applicable.

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
