# Peer review of "Chemical Characterization of Plant Extracts and Evaluation of their Nematicidal and Phytotoxic Potential"

_molecules, 2021, doi:10.3390/molecules26082216_

Round 1

Reviewer 1 Report

The manuscript is well prepared and novel. It has merit for publication in Molecules journal. I have some limited comments that if addressed will improve the manuscript prior to acceptance. 

1) For the non-expert: please explain the use of fluopyrgam and abamectin as positive controls. 

2) Page 7, line 183. Do the authors have any further indication on these impurities? Based on the capabilities of the research group i think that more specific data could be provided. Same page, line 203, this sentence needs better connection to the rest of this paragraph's content. 

3) Page 8, lines 216-217. The authors refer to synergistic effects among sterols and flavonoids. I think more discussion can appear, since such effects are mentioned in the literature, and extended to other classes as well (e.g., aglycones with glycosides). Conjugates of sterols can also be present. 

4) Minor comments: Page 2, line 69, identify not "identified". Page 7, line 177, nematicidal is preferable (consider it also in the keywords section). Page 8, line 240, shifts not "shifted". Page 9, line 256, DMSO-d6 not "DMSOd6". Page 11, line 312, please remove underlining in Celsius symbol. In the introduction, lines 50-51, concerning natural products activity against phytopathogenic nematodes authors can add other references on this topic such as: Ntalli et al., 2020, "Nematicidal Activity and Phytochemistry of Greek Lamiaceae Species", in Agronomy, 10(8), 1119, Ntalli et al., "Nematicidal Activity of Stevia rebaudiana (Bertoni) Assisted by Phytochemical Analysis", Toxins, 2020, 12, 319.

Reviewer 2 Report

The manuscript submitted for review concerns an interesting issue. However, in my opinion, it still needs a major revision.

  1. At the end of the abstract, please write a final (general) conclusion from your research.
  2. It is necessary to organize the manuscript, especially this note concerns materials and methods (duplicate content is present).
  3. Please let me know if negative tests consisting of the extraction medium (eg DMSO with TWEEN 20) were performed and the influence of such tests on nematodes and plants was investigated? It is necessary to present the results of such observations.

I would also like to ask the authors, in response to the review, to present their point of view, which shows the influence of the tested compounds on nematodes and plants. Writing precisely which potential metabolic pathways in nematodes and plants can be affected during treatment? Do the tested substances penetrate the body of the nematodes, or only surface action, and if so, what is the surface effect of these compounds? I believe that these types of views should be included in the Discussion.

Round 2

Reviewer 1 Report

The authors managed to address all comments raised. I suggest acceptance of the manuscript. 

Reviewer 2 Report

The authors exhaustively commented on my comments and suggestions. As it stands, the manuscript is much better and deserves to be published.